# Activation Matters: Adaptive and Activated Negative Labels for OOD Detection with Vision-Language Models

## Abstract

Out-of-distribution (OOD) detection aims to identify samples that deviate from in-distribution (ID). One popular pipeline addresses this by introducing negative labels distant from ID classes and detecting OOD based on their distance to these labels. However, such labels may present poor activation on OOD samples, failing to capture the OOD characteristics. To address this, we propose an Adaptive and Activated Negative labels guided approach (AANeg), which dynamically evaluates activation levels across the corpus dataset and selects words with high activation responses as negative labels. Specifically, AANeg identifies high-confidence test images online and accumulates their assignment probabilities over the corpus to construct a label activation metric. Such a metric leverages historical test samples to adaptively align with the test distribution, enabling the selection of distribution-adaptive activated negative labels. By further exploring the activation information within the current testing batch, we introduce a more fine-grained, batch-adaptive variant. To fully utilize label activation knowledge, we propose an activation-aware score function that emphasizes negative labels with stronger activations, boosting performance and enhancing its robustness to the label number. Our approach is zero-shot, training-free, test-efficient, highly scalable, and grounded in theoretical justification. Notably, on the large-scale ImageNet benchmark, AANeg significantly reduces the FPR95 from 17.5% to 9.8%. Codes will be released.

## 1 Introduction

In open environments, artificial intelligence (AI) models inevitably encounter out-of-distribution (OOD) data, *i.e.*, samples outside predefined categories. Existing vision models often misclassify these OOD samples as known categories (Nguyen et al., 2015), posing significant safety issues. Therefore, accurately detecting OOD samples is critical for deploying safe AI models.

Traditional OOD detection methods in the image domain primarily rely on visual features (Hendrycks & Gimpel, 2016; Lee et al., 2018; Liu et al., 2020). Recently, with the rapid development of vision-language models, leveraging textual knowledge to enhance OOD detection has gained increasing attention (Ming et al., 2022a; Wang et al., 2023; Esmaeilpour et al., 2022). Among these, NegLabel (Jiang et al., 2024) mines thousands of negative labels by identifying words with large cosine distances from in-distribution (ID) labels and detects OOD images by selecting those with higher cosine similarity to negative labels. Although NegLabel achieves impressive results, many negative labels present poor activation on a certain OOD test set, as shown in Fig. 1a. This limits their effectiveness and introduces noise, as these labels are closer to ID data and fail to capture the characteristics of OOD images effectively. Removing these less activated labels benefits the detection of OOD samples, as illustrated in Fig. 1b, highlighting their negative impact. This motivates us to explore negative labels with stronger activation on OOD samples to enhance OOD detection.

To this end, we propose an Adaptive and Activated Negative labels guided approach (AANeg) for OOD detection. At the core of AANeg is an activation metric that quantifies how "active" a particular class is across a dataset, measured by the average classification probability of a label across the associated images, as defined in Eq. 5. Using this metric, we dynamically evaluate the activation of each class in the corpus set across ID and OOD datasets, approximated using high-confidence positive

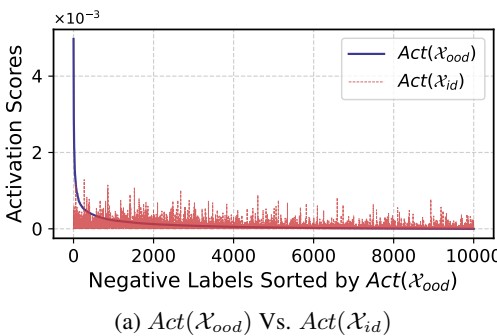
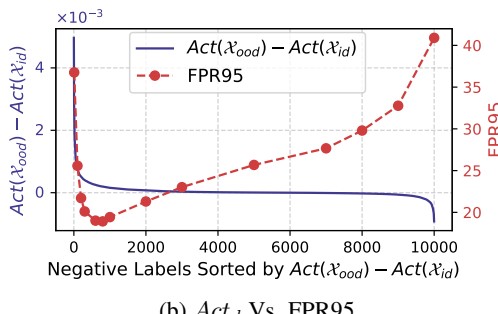

(a) $Act(\mathcal{X}_{ood})$ Vs. $Act(\mathcal{X}_{id})$

(b) $Act_d$ Vs. FPR95

Figure 1: Activation analyses with negative labels mined in (Jiang et al., 2024). (a) Negative labels on a specific OOD dataset exhibit a long-tailed activation score distribution. Some labels activate more strongly on the ID dataset than on OOD, potentially misleading OOD detection. (2) A small subset of negative labels strongly activates on OOD, enabling effective detection. Most labels respond similarly across ID and OOD, slightly harming detection, while some activate higher on ID, significantly degrading performance. The FPR95 results are obtained with negative labels of top activations via Eq. 4. These analyses use ground truth labels from ImageNet (ID) and Places (OOD) datasets.

and negative test images. This activation information captures the overall characteristics of the test distribution, enabling the selection of distribution-adaptive activated labels, which present high activation on negative images and low activation on positive images. Additionally, we explore the activation information within the current testing batch, resulting in a more fine-grained, batch-adaptive variant. To ensure stability at the beginning of testing, we initialize the activation scores using ID labels and noise images as positive and negative samples, respectively. Furthermore, considering the varying importance of different negative labels, as validated by their activation levels, we introduce an activation-aware scoring function for OOD detection that emphasizes labels with higher activation. This score function not only boosts the OOD detection performance but also enhances its robustness to the label number.

Extensive experiments are conducted to validate the effectiveness of our proposed methods. On the large-scale ImageNet dataset, our method reduces the FPR95 of the activation-agnostic NegLabel (Jiang et al., 2024) by 15.6% and outperforms the current leading approach (Chen et al., 2024) by 7.7%. Unlike existing methods (Jiang et al., 2024; Zhang & Zhang, 2024), which typically introduce thousands of negative labels to cover activated labels but inevitably include less activated ones, our method specifically targets activated labels, achieving superior performance with a significantly reduced number of labels. Moreover, our approach is zero-shot, training-free, and test-efficient, demonstrating high scalability to different model backbones and robustness to near-OOD, full-spectrum OOD, and medical OOD settings. Theoretical analysis further explains its effectiveness. We summarize our contributions as follows:

- We introduce an activation metric to quantify how "active" a particular class is across a dataset and reveal that current methods employ less-activated negative labels for OOD datasets, which hinders the distinction of OOD samples. This motivates us to explore activated negative labels to enhance OOD detection.

- To this end, we explore activated negative labels by dynamically estimating the activation scores across the entire corpus dataset. We introduce distribution-adaptive and batch-adaptive variants and a novel activation-aware scoring function to fully utilize the mined activation knowledge. Theoretical analysis is conducted to explain its effectiveness.

- We conduct extensive experiments to validate the proposed components. Our AANeg outperforms the leading method by 7.7% in FPR95 on the large-scale ImageNet benchmark. Our approach is zero-shot, training-free, test-efficient, and presents high scalability to different backbones and diverse task settings.

## 2 RELATED WORK

**OOD Detection** aims to identify test samples with undefined semantic concepts, thereby enhancing the reliability of models in open environments. Classical OOD detection methods typically explore knowledge exclusively from the visual domain and can be roughly categorized into score-based (Hendrycks & Gimpel, 2016; Lee et al., 2018; Liang et al., 2017; Liu et al., 2020; Wang et al., 2021; Huang & Li, 2021; Wang et al., 2022; Wei et al., 2022; Sun et al., 2021), distance-based (Tack et al., 2020; Tao et al., 2023; Sun et al., 2022; Du et al., 2022a; Ming et al., 2022b; Sehwag et al., 2021), and generative-based (Ryu et al., 2018; Kong & Ramanan, 2021) approaches. Among these, score-based methods are the most popular, including score functions based on prediction confidence (Hendrycks & Gimpel, 2016; Liang et al., 2017; Sun et al., 2021; Wang et al., 2022; Wei et al., 2022), additional discriminators (Kong & Ramanan, 2021), and energy score (Liu et al., 2020; Wang et al., 2021).

Recently, with the rise of vision-language models (Radford et al., 2021; Jia et al., 2021; Zhai et al., 2023), enhancing visual OOD detection by leveraging text knowledge has gained increasing attention. ZOC (Esmaeilpour et al., 2022) pioneered this direction by generating potential OOD labels with a learned captioner. Adapting the text branch with prompt tuning is a popular approach, where methods such as LoCoOp (Miyai et al., 2024), LAPT (Zhang et al., 2024b), CLIPN (Wang et al., 2023), and LSN (Nie et al., 2024) introduce negative training samples via background features, image generation, additional training sets, and local croppings, respectively. Beyond single-text modality fine-tuning, multi-modal fine-tuning has been explored in (Kim & Hwang, 2025) by jointly tuning both the text and visual branches.

Unlike these tuning-based methods, which typically require labeled training data, some approaches focus on training-free zero-shot pipelines. For instance, MCM (Ming et al., 2022a) detects OOD samples by analyzing the similarity distribution between test images and ID labels. EOE (Cao et al., 2024), NegLabel (Jiang et al., 2024), and CSP (Chen et al., 2024) further advance this by introducing negative labels and utilizing the similarity between test images and ID/negative labels to enhance OOD detection. While these methods achieve good results, we reveal that many of these negative labels present poor activation on OOD images, hindering the accurate detection of OOD samples, as shown in Fig. 1. To address this issue, we explore more activated negative labels and design a corresponding activation-aware score function, significantly reducing the FPR95 on the ImageNet dataset from 25.4%, achieved by the NegLabel, to 9.8%.

**Test Time Adaptation** enables models to dynamically update during testing to adapt to changes in test data (Liang et al., 2023), which has been recently introduced into OOD detection. Some methods (Gao et al., 2023; Yang et al., 2023b; Fan et al., 2024) require test-time optimization, which, although achieving certain performance improvements, significantly reduces testing speed. Recently, some methods (Zhang & Zhang, 2024; Yang et al., 2025) have also sought to avoid test-time optimization, achieving rapid adaptation to the environment. For example, OODD (Yang et al., 2025) caches high-confidence positive/negative samples in a dictionary and detects OOD samples based on the cosine similarity between the test samples and the cached features. The most similar method is AdaNeg (Zhang & Zhang, 2024), which also introduces adaptive negative proxies. The difference is that AdaNeg primarily revises the activated negative labels while retaining these less-activated ones. In contrast, we reduce the influence of less-activated labels through a novel activation-guided label mining strategy and a corresponding activation-aware score, significantly boosting OOD detection.

## 3 METHODS

### 3.1 PRELIMINARIES

**OOD Detection.** Consider $\mathcal{X}$ as the image domain and $\mathcal{Y}^+ = \{y_1, \ldots, y_C\}$ as the set of ID class labels, where $\mathcal{Y}^+$ consists of textual elements such as $\mathcal{Y}^+ = \{\text{cat}, \text{dog}, \ldots, \text{bird}\}$, and $C$ is the total number of classes. Let $\boldsymbol{x}^{in}$ and $\boldsymbol{x}^{ood}$ represent random variables corresponding to ID and OOD samples from $\mathcal{X}$, respectively. The marginal distributions for ID and OOD samples are denoted by $\mathcal{P}_{\boldsymbol{x}}^{in}$ and $\mathcal{P}_{\boldsymbol{x}}^{ood}$. In standard classification tasks, it is assumed that a test image $\boldsymbol{x}$ belongs to the ID distribution and is associated with a specific ID label, *i.e.*, $\boldsymbol{x} \in \mathcal{P}_{\boldsymbol{x}}^{in}$ and $y \in \mathcal{Y}^+$, where $y$ is the label of $\boldsymbol{x}$. However, in open environments, AI systems often encounter data from unknown classes, characterized by $\boldsymbol{x} \in \mathcal{P}_{\boldsymbol{x}}^{ood}$ and $y \notin \mathcal{Y}^+$. These OOD samples are typically misclassified as the known ID categories (Scheirer et al., 2012; Nguyen et al., 2015), potentially resulting in unsafe

decisions. To tackle such issues, OOD detection aims to reliably classify ID samples into their respective categories while rejecting OOD samples as non-ID. Classification among ID categories is done using a $C$-way classifier, following conventional methods (Krizhevsky et al., 2012; He et al., 2016). Meanwhile, OOD detection employs a scoring function $S$ (Lee et al., 2018; Liang et al., 2017; Liu et al., 2020) to distinguish between ID and OOD inputs:

$$G_\gamma(\boldsymbol{x}) = \begin{cases} \text{ID}, & \text{if } S(\boldsymbol{x}) \geq \gamma; \\ \text{OOD}, & \text{otherwise,} \end{cases} \tag{1}$$

where $G_\gamma$ is the OOD detector with a threshold $\gamma \in \mathcal{R}$ and $S(\boldsymbol{x})$ is a scoring function assigning higher scores to samples likely belonging to ID classes.

**CLIP and NegLabel.** For an ID test image $\boldsymbol{x}$ belonging to the label space $\mathcal{Y}^+$, we extract its image feature vector $\boldsymbol{v} = f_{img}(\boldsymbol{x}) \in \mathcal{R}^D$ and the text feature $\boldsymbol{t}_i = f_{txt}(\rho(y_i)) \in \mathcal{R}^D$ using pre-trained CLIP encoders, where $D$ denotes the feature dimension. The functions $f_{img}(\cdot)$ and $f_{txt}(\cdot)$ represent images and text encoders, respectively. The function $\rho(\cdot)$ serves as a text prompt mechanism, typically defined as 'a photo of a <label>,' where '<label>' corresponds to the actual class name such as 'cat' or 'dog'. Both $\boldsymbol{v}$ and $\boldsymbol{t}_i$ are normalized using $L_2$ normalization along the dimension $D$. The zero-shot classification probabilities are then computed with the similarity between $\boldsymbol{v}$ and $\boldsymbol{t}_i$:

$$\boldsymbol{p}_i^{id} = \frac{\exp(\boldsymbol{v}\boldsymbol{t}_i/\tau)}{\sum_{j=1}^C \exp(\boldsymbol{v}\boldsymbol{t}_j/\tau)}, \tag{2}$$

where $\tau > 0$ is the temperature scaling factor.

The CLIP model has been recently extended to OOD detection (Ming et al., 2022a; Jiang et al., 2024; Zhang et al., 2024b). Specifically, Jiang *et al.* (Jiang et al., 2024) introduce negative class labels $\mathcal{Y}^- = \{\widetilde{y}_1, \ldots, \widetilde{y}_M\}$ by mining text labels distant from ID classes $\mathcal{Y}^+$ within an extensive corpus dataset $\mathcal{Y}^{cor} = \{\widehat{y}_1, \ldots, \widehat{y}_N\}$:

$$\mathcal{Y}^- = Top(\{d_i\}_{i=1}^N, \mathcal{Y}^{cor}, M) \tag{3}$$

where $d_i$ measures the cosine distance between $\widehat{y}_i$ and ID label set $\mathcal{Y}^+$. $M$ and $N$ respectively denote the number of selected negative classes and all classes in the corpus dataset, and $N \geq M$. The operation $TOP(A, B, M)$ retrieves the indices of the top-M largest elements in set $A$ and uses them to select the corresponding elements from set $B$. Sets $\mathcal{Y}^-$ and $\mathcal{Y}^+$ are disjoint, *i.e.*, $\mathcal{Y}^- \cap \mathcal{Y}^+ = \emptyset$. Then, the ID images can be detected as those with higher similarity to ID labels and lower similarity to negative ones, leading to the following score function:

$$S_{nl}(\boldsymbol{v}) = \sum_{i=1}^C \frac{\exp(\boldsymbol{v}\boldsymbol{t}_i/\tau)}{\sum_{j=1}^C \exp(\boldsymbol{v}\boldsymbol{t}_j/\tau) + \sum_{j=1}^M \exp(\boldsymbol{v}\widetilde{\boldsymbol{t}}_j/\tau)}, \tag{4}$$

where $\widetilde{\boldsymbol{t}}_i = f_{txt}(\rho(\widetilde{y}_i)) \in \mathcal{R}^D$ is the text feature of mined negative label $\widetilde{y}_i$.

## 3.2 MOTIVATION: LABEL ACTIVATION ANALYSES

Although the negative labels described above perform well, they have a significant limitation that hinders their effectiveness. Specifically, the negative labels in Eq. 3 are derived solely from ID labels, without considering the test distribution in real-world applications. Consequently, many negative labels exhibit very low activation scores for a specific OOD test set and hinder the OOD detection, as shown in Fig. 1. To better understand this, we first introduce the concept of the activation score, which measures the average probability assignment of class $\widehat{y}_i$ on a dataset $\mathcal{X}$:

$$Act(\mathcal{X}, \widehat{y}_i) = \frac{1}{|\mathcal{X}|} \sum_{\boldsymbol{x} \in \mathcal{X}} \frac{\exp(\boldsymbol{v}\widehat{\boldsymbol{t}}_i/\tau)}{\sum_{j=1}^C \exp(\boldsymbol{v}\boldsymbol{t}_j/\tau) + \sum_{j=1}^N \exp(\boldsymbol{v}\widehat{\boldsymbol{t}}_j/\tau)}, \tag{5}$$

where $\widehat{\boldsymbol{t}}_i = f_{txt}(\rho(\widehat{y}_i)) \in \mathcal{R}^D$. The $Act(\mathcal{X}, \widehat{y}_i)$ reflects the average similarity between the class $\widehat{y}_i$ and images in the dataset $\mathcal{X}$. A higher $Act(\mathcal{X}, \widehat{y}_i)$ indicates greater similarity between the samples in $\mathcal{X}$ and class $\widehat{y}_i$, and vice versa. An ideal negative label $\widehat{y}_i$ should exhibit higher activation scores on OOD dataset (*e.g.*, higher $Act(\mathcal{X}_{ood}, \widehat{y}_i)$) and lower activation scores on ID dataset (*e.g.*, lower $Act(\mathcal{X}_{id}, \widehat{y}_i)$). However, as shown in Fig. 1a, although most negative labels derived in Eq. 3 exhibit low activation scores on the ID dataset, many simultaneously exhibit low activation scores on the OOD dataset—sometimes even lower than those observed on the ID dataset. These negative labels, which demonstrate lower activation scores on OOD samples, adversely impact OOD detection, as evidenced in Fig. 1b.

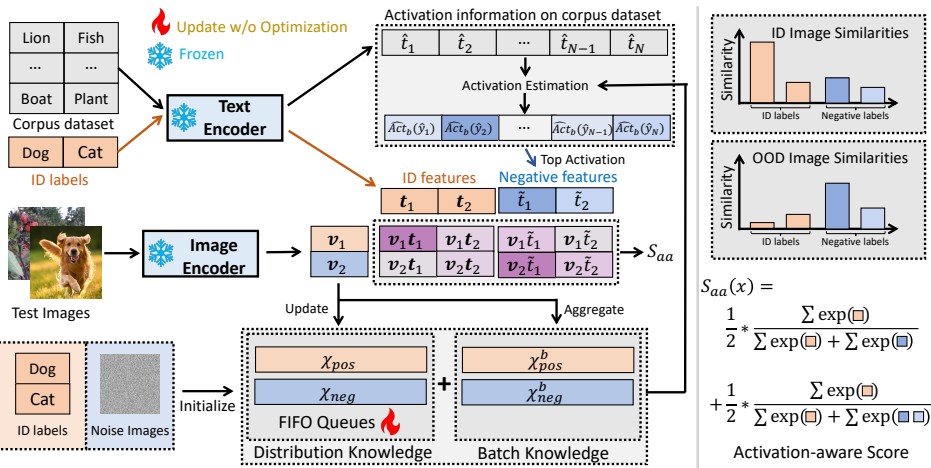

Figure 2: Overall framework of AANeg. We dynamically explore adaptive and activated negative labels from the corpus dataset, where the activation information is measured based on the similarity between texts and the mined positive/negative images. The activation-aware score is illustrated as a simplified example of Eq. 15 with $M = 2$ and $C = 2$.

### 3.3 OUR APPROACH

**Distribution-adaptive Activated Labels.** The above analyses on label activation motivate us to explore negative labels with higher activation on the OOD dataset and lower activation on the ID dataset, *e.g.*, to find the negative labels that present high activation scores for OOD detection:

$$\mathcal{Y}^- = Top(\{Act_d(\widehat{y}_i)\}_{i=1}^N, \mathcal{Y}^{cor}, M), \tag{6}$$

$$\text{where} \quad Act_d(\widehat{y}_i) = Act(\mathcal{X}_{ood}, \widehat{y}_i) - Act(\mathcal{X}_{id}, \widehat{y}_i). \tag{7}$$

However, in open environments, OOD data is generally unknown and may even change dynamically, making it difficult to obtain in advance. To address these problems, we propose a test-time adaptation strategy to dynamically evaluate the activation levels of candidate labels online, thereby selecting the most effective negative labels. Specifically, we approximate the activation score in Eq. 7 with cached positive and negative images:

$$\widehat{Act_d}(\widehat{y}_i) = Act(\mathcal{X}_{neg}, \widehat{y}_i) - Act(\mathcal{X}_{pos}, \widehat{y}_i), \tag{8}$$

where $\mathcal{X}_{neg}$ and $\mathcal{X}_{pos}$ are fixed-length first-in-first-out (FIFO) queues that are dynamically updated with high-confidence positive and negative samples, respectively:

$$
\begin{aligned}
\mathcal{X}_{pos} &= \text{QueueUpdate}(\mathcal{X}_{pos}, \boldsymbol{v} \in \mathcal{B} \mid S_{aa}(\boldsymbol{v}) \geq \gamma + (1-\gamma)g, L), \\
\mathcal{X}_{neg} &= \text{QueueUpdate}(\mathcal{X}_{neg}, \boldsymbol{v} \in \mathcal{B} \mid S_{aa}(\boldsymbol{v}) < \gamma - \gamma g, L),
\end{aligned}
\tag{9}
$$

where $\gamma \in [0, 1]$ is the threshold distinguishing ID and OOD samples, $g \in [0, 1]$ is the gap used to filter high-confidence samples, $\mathcal{B}$ indicates the test image batch, and $L$ is the capacity of the queue. $S_{aa}$ is our proposed activation-aware score function, as defined in Eq. 15. We construct $\mathcal{X}_{pos/neg}$ with image features $\boldsymbol{v}$, equivalent to vanilla images and more compact. This FIFO queue design ensures that the activation score remains responsive to environmental changes by focusing on the most recent high-confidence samples. It effectively filters outdated information, allowing the model to adapt dynamically and maintain robust performance in evolving scenarios.

To ensure a stable start for our method during testing, we respectively initialize $\mathcal{X}_{pos}$ and $\mathcal{X}_{neg}$ with features of ID labels and noise images:

$$
\begin{aligned}
\mathcal{X}_{pos} &= \text{Sampling}(\{\boldsymbol{t}_i\}_{i=1}^C, L) \\
\mathcal{X}_{neg} &= \{f_{img}(\boldsymbol{x}_i^{noise})\}_{i=1}^L,
\end{aligned}
\tag{10}
$$

where $\boldsymbol{x}_i^{noise}$ is the image with random Gaussian noises and $\text{Sampling}(A, L)$ represents the operation of randomly selecting $L$ elements from set $A$. We initialize the positive image set $\mathcal{X}_{pos}$ using ID label

features, considering the alignment between text and image features in the shared feature space. This initialization helps construct the initial activated labels and the corresponding $S_{aa}$ score function, providing a solid foundation for subsequent updates.

**Batch-Adaptive Activated Labels.** The above method enhances OOD detection by utilizing activation information from historical test samples. However, in addition to historical samples, the activation information in the current testing batch is also critical, as it captures the self-characteristics of test instances, especially under scenarios with temporal shifts. To address this, we propose a more fine-grained, batch-adaptive variant by incorporating the activation information within the current testing batch:

$$
\begin{aligned}
\mathcal{X}^b_{pos} &= \{\boldsymbol{v} \in \mathcal{B} \mid S_{aa}(\boldsymbol{v}) \geq \gamma + (1-\gamma)g\}, \\
\mathcal{X}^b_{neg} &= \{\boldsymbol{v} \in \mathcal{B} \mid S_{aa}(\boldsymbol{v}) < \gamma - \gamma g\},
\end{aligned}
\tag{11}
$$

where $\mathcal{X}^b_{pos}$ and $\mathcal{X}^b_{neg}$ represent the possible positive and negative samples in the testing batch $\mathcal{B}$. We then define the batch-adaptive activation score as:

$$
\widehat{Act}_b(\widehat{y}_i) = Act_b(\mathcal{X}_{neg}, \widehat{y}_i) - Act_b(\mathcal{X}_{pos}, \widehat{y}_i),
\tag{12}
$$

where

$$
Act_b(\mathcal{X}_{pos}, \widehat{y}_i) = \begin{cases} \alpha Act(\mathcal{X}_{pos}, \widehat{y}_i) + (1-\alpha)Act(\mathcal{X}^b_{pos}, \widehat{y}_i) & \text{if } |\mathcal{X}^b_{pos}| > 0 \\ Act(\mathcal{X}_{pos}, \widehat{y}_i) & \text{if } |\mathcal{X}^b_{pos}| = 0 \end{cases}
\tag{13}
$$

$$
Act_b(\mathcal{X}_{neg}, \widehat{y}_i) = \begin{cases} \alpha Act(\mathcal{X}_{neg}, \widehat{y}_i) + (1-\alpha)Act(\mathcal{X}^b_{neg}, \widehat{y}_i) & \text{if } |\mathcal{X}^b_{neg}| > 0 \\ Act(\mathcal{X}_{neg}, \widehat{y}_i) & \text{if } |\mathcal{X}^b_{neg}| = 0 \end{cases}
\tag{14}
$$

where $\alpha \in [0, 1]$ balances activation information from historical samples and the current batch, and $|\cdot|$ measures the set size.

**Activation-aware Score.** Considering that the activation scores of different negative labels vary, *i.e.*, their importance differs, we introduce for OOD detection the following score to implicitly assign higher weights to those with higher activation scores:

$$
S_{aa}(\boldsymbol{v}) = \frac{1}{M} \sum_{m=1}^{M} \sum_{i=1}^{C} \frac{\exp(\boldsymbol{v}\boldsymbol{t}_i/\tau)}{\sum_{j=1}^{C} \exp(\boldsymbol{v}\boldsymbol{t}_j/\tau) + \sum_{j=1}^{m} \exp(\boldsymbol{v}\widetilde{\boldsymbol{t}}_j/\tau)}.
\tag{15}
$$

Recalling that the activated labels are selected based on ranked activation scores in Eq. 6, the ranking ensures that $\widehat{Act}_b(\widetilde{y}_i) \geq \widehat{Act}_b(\widetilde{y}_j)$ if $i < j$. This guarantees that labels with stronger activation (*e.g.*, $\widetilde{\boldsymbol{t}}_j$ with smaller $j$) dominate the score, as their contributions are amplified by their repeated occurrence in the denominator. We find that this method not only improves OOD detection performance but also significantly enhances its robustness to the number of negative labels, as verified in Fig. 3a. We also observe that Equations (15) and (9) exhibit a mutual enhancement: the positive/negative images selected by Eq. (9) enable more accurate estimation of label activation information, providing more effective negative labels for the score calculation in Eq. (15). In turn, the improved OOD detection capability of Eq. (15) further facilitates the distinguishing of positive and negative samples in Eq. (9), as analyzed in Tab. A14. The overall framework is shown in Fig. 2 and summarized in Algorithm A1.

**Theoretical Insight.** We follow (Jiang et al., 2024) to conduct the theoretical analysis from the perspective of multi-label classification and model the partial derivative of $FPR_\lambda$ with respect to the number $M$ of negative labels as:

$$
\frac{\partial FPR_\lambda}{\partial M} = \frac{1}{2} \cdot \frac{\partial erf(z)}{\partial z} \cdot \frac{\partial z}{\partial M} = \frac{e^{-z^2}}{2\sqrt{2\pi}} \cdot \frac{p_1 - p_2}{\sqrt{Mp_2(1-p_2)}},
\tag{16}
$$

where $p_1 = P(sim(\boldsymbol{x}_{id}, \widetilde{y}_i) \geq \psi | f, \boldsymbol{x}_{id}, \widetilde{y}_i)$ indicates the average probability of classifying the ID image $\boldsymbol{x}_{id}$ as the negative label $\widetilde{y}_i$, $sim(\boldsymbol{x}_{id}, \widetilde{y}_i)$ represents the similarity between $\boldsymbol{x}_{id}$ and $\widetilde{y}_i$, and $\psi$ is a threshold. $p_2$ is similarly defined with the OOD image $\boldsymbol{x}_{ood}$. $erf(z) = \frac{2}{\sqrt{\pi}} \int_0^z e^{-t^2} dt$ is the error function and $z = \sqrt{\frac{p_1(1-p_1)}{p_2(1-p_2)}} erf^{-1}(2\lambda - 1) + \frac{\sqrt{M}(p_1-p_2)}{\sqrt{2p_2(1-p_2)}}$. As shown in Eq. 16, to reduce the $FPR_\lambda$ by increasing the number $M$ of negative labels, a prerequisite is that $p_1 - p_2 < 0$, suggesting

Table 1: OOD detection results with ImageNet-1k, where a VITB/16 CLIP encoder is adopted.

| Methods | OOD datasets | | | | | | | | | |
| | INaturalist | | Sun | | Places | | Textures | | Average | |
| | AUROC↑ | FPR95↓ | AUROC↑ | FPR95↓ | AUROC↑ | FPR95↓ | AUROC↑ | FPR95↓ | AUROC↑ | FPR95↓ |
| --- | --- | --- | --- | --- | --- | --- | --- | --- | --- | --- |
| **Methods requiring training (or fine-tuning)** | | | | | | | | | | |
| ZOC (Esmaeilpour et al., 2022) | 86.09 | 87.30 | 81.20 | 81.51 | 83.39 | 73.06 | 76.46 | 98.90 | 81.79 | 85.19 |
| LSN (Nie et al., 2024) | 95.83 | 21.56 | 94.35 | 26.32 | 91.25 | 34.48 | 90.42 | 38.54 | 92.96 | 30.22 |
| CLIPN (Wang et al., 2023) | 95.27 | 23.94 | 93.93 | 26.17 | 92.28 | 33.45 | 90.93 | 40.83 | 93.10 | 31.10 |
| LoCoOp (Miyai et al., 2024) | 96.86 | 16.05 | 95.07 | 23.44 | 91.98 | 32.87 | 90.19 | 42.28 | 93.52 | 28.66 |
| NegPrompt (Li et al., 2024) | 98.73 | 6.32 | 95.55 | 22.89 | 93.34 | 27.60 | 91.60 | 35.21 | 94.81 | 23.01 |
| CMA (Kim & Hwang, 2025) | 99.62 | 1.65 | 96.36 | 16.84 | 93.11 | 27.65 | 91.64 | 33.58 | 95.13 | 19.93 |
| **Zero-Shot Training-free Methods** | | | | | | | | | | |
| MCM (Ming et al., 2022a) | 94.59 | 32.20 | 92.25 | 38.80 | 90.31 | 46.20 | 86.12 | 58.50 | 90.82 | 43.93 |
| CoVer (Zhang et al., 2024a) | 95.98 | 22.55 | 93.42 | 32.85 | 90.27 | 40.71 | 90.14 | 43.39 | 92.45 | 34.88 |
| Lee *et al.* (Lee et al., 2025) | 96.89 | 23.84 | 93.69 | 30.11 | 93.17 | 29.86 | 88.47 | 47.35 | 93.05 | 32.79 |
| EOE (Cao et al., 2024) | 97.52 | 12.29 | 95.73 | 20.04 | 92.95 | 30.16 | 85.64 | 57.53 | 92.96 | 30.09 |
| NegLabel (Jiang et al., 2024) | 99.49 | 1.91 | 95.49 | 20.53 | 91.64 | 35.59 | 90.22 | 43.56 | 94.21 | 25.40 |
| AdaNeg (Zhang & Zhang, 2024) | 99.71 | 0.59 | 97.44 | 9.50 | 94.55 | 34.34 | 94.93 | 31.27 | 96.66 | 18.92 |
| OODD (Yang et al., 2025) | 99.79 | 0.85 | 97.17 | 12.94 | 92.51 | 30.68 | 94.51 | 30.67 | 96.00 | 18.79 |
| CSP (Chen et al., 2024) | 99.60 | 1.54 | 96.66 | 13.66 | 92.90 | 29.32 | 93.86 | 25.52 | 95.76 | 17.51 |
| **AANeg (Ours)** | **99.84** | **0.42** | **99.07** | **3.53** | **95.87** | **21.90** | **97.11** | **13.38** | **97.97** | **9.81** |

that $sim(\boldsymbol{x}_{id}, \widetilde{y}_i) < sim(\boldsymbol{x}_{ood}, \widetilde{y}_i)$ holds on average. In other words, the negative label $\widetilde{y}_i$ should have higher similarity with the OOD samples and lower similarity with the ID samples.

There is a close relationship between the theoretical objective in Eq. 16 and our algorithm implementation in Eq. 7. Specifically, $Act(\mathcal{X}_{ood}, \widehat{y}_i)$ measures the activation level of candidate label $\widehat{y}_i$ on the OOD dataset by averaging the normalized similarity between $\widehat{y}_i$ and OOD images; similarly, $Act(\mathcal{X}_{id}, \widehat{y}_i)$ reflects the normalized similarity between $\widehat{y}_i$ and ID images. According to Eq. 6, we select the ideal negative labels $\{\widetilde{y}_i\}_{i=1}^{M}$ that exhibit higher activation with OOD samples while maintaining lower activation with ID samples, explicitly ensuring $p_1 - p_2 < 0$ in Eq. 16.

# 4 EXPERIMENTS

## 4.1 SETUP

**Datasets.** We primarily conducted experiments using ImageNet-1k (Deng et al., 2009) as the ID dataset. We adopted four OOD datasets (Van Horn et al., 2018; Xiao et al., 2010; Zhou et al., 2017; Cimpoi et al., 2014) following common practice (Huang & Li, 2021; Ming et al., 2022a; Jiang et al., 2024) and also performed experiments under the OpenOOD setting (Zhang et al., 2023; Yang et al., 2022). Additionally, we validated our method under the full-spectrum setting (Yang et al., 2023a), with a smaller ID dataset of CIFAR (Krizhevsky et al., 2009), and with medical images (Vayá et al., 2020; Zhang et al., 2023). More details are provided in Appendix A6.2.

**Implementation Details.** We employ the visual encoder of VITB/16, pretrained with CLIP (Radford et al., 2021), and additionally investigate other backbone architectures in Tab. A12. Following the design of NegLabel, we use the text prompt "The nice ¡label¿" and set the temperature parameter $\tau$ to 0.01. In our method, we set the number of negative labels to $M = 1000$ in Eq. 15, the gap $g = 0.2$, and $L = 300$ in Eq. 9, and $\alpha = 0.95$ in Eq. 12. As shown in Fig. A4b, an automatically determined threshold $\gamma$ generally performs comparably to manually searched ones. We adopt the evaluation metrics of FPR95, AUROC, and ID ACC, following standard protocols (Huang & Li, 2021; Ming et al., 2022a; Jiang et al., 2024). All experiments are conducted with an NVIDIA H100 GPU.

## 4.2 MAIN RESULTS

**ImageNet Results.** As shown in Tab. 1, our AANeg significantly outperforms existing training-free methods and even surpasses techniques that require additional training. Detailed discussions are provided in Sec. A6.3 and complete comparisons are presented in Tab. A6.

**ImageNet Results with OpenOOD Setup.** As shown in Tab. 2, AANeg surpasses existing zero-shot training-free methods and achieves performance comparable to training-required competitors. Notably, it outperforms the close competitors (Jiang et al., 2024; Zhang & Zhang, 2024), especially

Table 2: OOD detection results under the OpenOOD setting, where ImageNet-1k is adopted as ID dataset. Full results are available in Tab. A7.

| Methods | FPR95 ↓ | | AUROC ↑ | | ACC ↑ |
|---|---|---|---|---|---|
| | Near-OOD | Far-OOD | Near-OOD | Far-OOD | ID |
| **Methods requiring training (or fine-tuning)** | | | | | |
| AugMix (Hendrycks et al., 2019b) + ReAct (Sun et al., 2021) | – | – | 79.94 | 93.70 | 77.63 |
| SCALE (Xu et al., 2023) | – | – | 81.36 | 96.53 | 76.18 |
| AugMix (Hendrycks et al., 2019b) + ASH (Djurisic et al., 2022) | – | – | 82.16 | 96.05 | 77.63 |
| LAPT (Zhang et al., 2024b) | 58.94 | 24.86 | 82.63 | 94.26 | 67.86 |
| CMA (Kim & Hwang, 2025) | 56.25 | 15.29 | 84.46 | 96.47 | 82.64 |
| **Zero-shot Training-free Methods** | | | | | |
| MCM (Ming et al., 2022a) | 79.02 | 68.54 | 60.11 | 84.77 | 66.28 |
| NegLabel (Jiang et al., 2024) | 69.45 | 23.73 | 75.18 | 94.85 | 66.82 |
| AdaNeg (Zhang & Zhang, 2024) | 67.51 | 17.31 | 76.70 | **96.43** | **67.13** |
| **AANeg (Ours)** | **60.06** | **17.21** | **84.53** | **96.43** | 66.82 |

Table 3: Full-spectrum OOD detection results under the OpenOOD setting, where ImageNet-1k, ImageNet-C, ImageNet-R, ImageNet-V2 are used as ID datasets. Full results are shown in Tab. A8.

| Methods | FPR95 ↓ | | AUROC ↑ | |
|---|---|---|---|---|
| | Near-OOD | Far-OOD | Near-OOD | Far-OOD |
| **Methods requiring training (or fine-tuning)** | | | | |
| AugMix (Hendrycks et al., 2019b) + SHE (Zhang et al., 2022) | 84.45 | 60.26 | 69.66 | 83.06 |
| LSA (Lu et al., 2023) | 70.56 | 48.06 | 78.22 | 86.85 |
| ISH + SCALE (Xu et al., 2023) | – | – | 68.04 | 89.46 |
| LAPT (Zhang et al., 2024b) | 71.18 | 33.07 | 74.77 | 92.14 |
| **Zero-shot Training-free Methods** | | | | |
| MCM (Ming et al., 2022a) | 85.37 | 69.87 | 58.97 | 77.11 |
| NegLabel (Jiang et al., 2024) | 76.25 | 33.30 | 72.77 | 92.02 |
| **AANeg (Ours)** | **68.71** | **22.48** | **78.90** | **94.35** |

in the challenging near-ood setting, demonstrating the effectiveness of activated negative labels. Our method also preserves ID classification accuracy by freezing the pre-trained CLIP model.

**Full-spectrum OOD Detection.** As shown in Tab. 3, our method not only demonstrates high distinguishability against semantic shifts but also exhibits strong robustness to covariate shifts.

**More results** on CIFAR and medical datasets are provided in Tab. A9 and Tab. A11, respectively.

### 4.3 ANALYSES AND DISCUSSIONS

**Ablation.** As shown in Tab. 4, adopting adaptive and activated negative labels significantly outperforms NegLabel, which uses fixed and activation-agnostic labels, verifying the importance of label activation. Additionally, incorporating current batch information brings a slight improvement and adopting activation-aware score consistently leads to an advantage.

**Label Number and $\mathcal{S}_{aa}$.** As illustrated in Fig. 3a, introducing adaptive and activated negative labels consistently outperforms NegLabel, validating the importance of label activation. Specifically, when the number of negative labels $M$ is small, the selected labels present higher activation scores, significantly reducing the OOD detection error. As $M$ increases, less-activated labels are gradually

Table 4: Ablation analyses, where results are reported with ImageNet ID dataset under the OpenOOD setup. "Dis-adapt", "Batch-adapt", and "AAScore" represent the distribution-adaptive activated score in Eq. 8, batch-adaptive variant in Eq. 11, and activation-aware score in Eq. 15, respectively.

| Dis-adapt | Batch-adapt | AAScore | Near-OOD FPR95 ↓ | Far-OOD FPR95 ↓ |
|---|---|---|---|---|
| | NegLabel Baseline | | 69.45 | 23.73 |
| ✓ | | | 61.57 | 17.52 |
| | ✓ | | 60.85 | 17.25 |
| ✓ | | ✓ | 60.59 | 17.27 |
| | ✓ | ✓ | 60.06 | 17.21 |

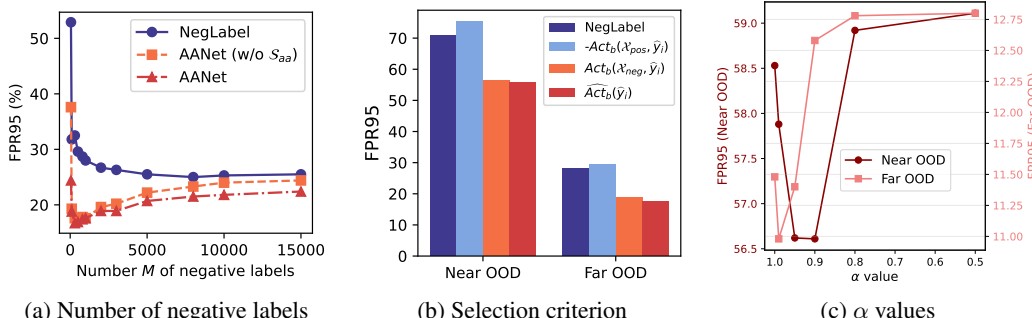

(a) Number of negative labels     (b) Selection criterion     (c) $\alpha$ values

Figure 3: Analyses on (a) number $M$ of selected negative labels, (b) selection criterion of negative labels, and (c) $\alpha$ values under OpenOOD setting.

Table 5: Time complexity analyses. 'Training' measures the training time, and 'Param.' presents the number of learnable parameters. 'FPS' reflects the inference speed with a batch size of 256.

| Methods | Backbones | Training | FPS ↑ | Param. | FPR95 ↓ |
|---|---|---|---|---|---|
| ZOC (Esmaeilpour et al., 2022) | | >24h | 287 | 336M | 85.19 |
| LoCoOp (Miyai et al., 2024) | | 9h | 625 | 8K | 28.66 |
| MCM (Ming et al., 2022a) | VITB/16 | – | 625 | – | 43.93 |
| NegLabel (Jiang et al., 2024) | | – | 592 | – | 25.40 |
| AdaNeg (Zhang & Zhang, 2024) | | – | 476 | – | 18.92 |
| **AANeg (Ours)** | | – | 527 | – | 9.81 |
| **AANeg (Ours)** | ResNet50 | – | 790 | – | 10.37 |

included, resulting in a gradual performance decline, consistent with the theoretical analysis in Sec. 3.3. Using the same score function $\mathcal{S}_{nl}$, our AANet (w/o $S_{aa}$) consistently outperforms NegLabel across different numbers of negative labels, fairly validating the advantage of our label selection strategy. Additionally, AANet consistently outperforms AANet (w/o $S_{aa}$), particularly when the number of negative labels is large. This confirms that incorporating label activation information into the score function enhances its robustness to the number of negative labels.

**Negative Label Selection Criterion.** As shown in Fig. 3b, mining negative labels with the activation information of negative samples significantly outperforms using positive information alone. Combining both positive and negative information yields the best results.

$\alpha$ **Analyses.** As shown in Fig. 3c, reducing the $\alpha$ values from 1.0 to 0.95 brings certain improvements, validating the effectiveness of batch knowledge. However, further decreasing the $\alpha$ value leads to overfitting to batch information, which in turn harms OOD identification. $\alpha = 0.95$ achieves a good balance between distribution and batch information and is adopted as the default setting.

**Complexity Analyses.** As shown in Table 5, our approach achieves superior performance with moderate test speed. We also observe that our method can achieve strong results even with smaller backbones (*e.g.*, ResNet50), offering advantages in both speed and accuracy.

**More analyses and discussions** on the queue length $L$, threshold $\gamma$, and gap $g$ in Eq. 9, different VLM backbones, various corpus datasets, batch size, activation metric, sample order, robustness to temporal shift, limitations, and visualization of activated labels are provided in Appendix A6.6.

## 5    CONCLUSION

In this paper, we proposed a novel OOD detection method that dynamically explored adaptive and activated negative labels during the test stage. We also designed an activation-aware score function to fully utilize the mined activation knowledge. Our approach was zero-shot, training-free, test-efficient, and grounded in theoretical justification. Additionally, it demonstrated high scalability to different model backbones and robustness to near-OOD, full-spectrum OOD, and medical OOD settings. We hope this work draws attention to the label activation information in the OOD detection community.

REPRODUCIBILITY STATEMENT

We have made significant efforts to ensure the reproducibility of our work. All datasets used in our experiments are publicly available. The methods and hyperparameter settings are thoroughly described in Section 4.1 and analyzed in Sections 4.3 and A6.6. Additionally, we will release all source code, scripts, and configuration files necessary to reproduce our results after the review process, ensuring that the experiments can be replicated easily.

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

## A6 APPENDIX

### A6.1 PSEUDO CODE

The pseudo code are illustrated in Algorithm A1.

---

**Algorithm A1** Adaptive and Activated Negative Labels for OOD Detection

---

**Require:** ID labels $\mathcal{Y}^+$, an external corpus dataset $\mathcal{Y}^{cor}$ and a test set $\mathcal{X}_{test}$
1: Constructing the FIFO queues $\mathcal{X}_{pos}$ and $\mathcal{X}_{neg}$, and initializing them with Eq. 10.
2: **for** Image batch $\mathcal{B} \in \mathcal{X}_{test}$ **do**
3:     Collecting positive set $\mathcal{X}_{pos}^b$ and negative set $\mathcal{X}_{neg}^b$ in batch $\mathcal{B}$ using Eq. 11
4:     Calculating the batch-adaptive activation score $\widehat{Act}_b(y_i)$ using Eq. 12
5:     Selecting the adaptive and activated negative labels $\mathcal{Y}^-$ using Eq. 6
6:     Generating the OOD detection score $S_{aa}$ using Eq. 15
7:     Updating the FIFO queues using Eq. 9
8: **end for**
9: **Return** Collected final scores $\{S_{aa}\}$

---

### A6.2 DATASET

We perform extensive experiments using the large-scale ImageNet-1k dataset (Deng et al., 2009) as the ID dataset. In line with previous works (Huang & Li, 2021; Ming et al., 2022a; Jiang et al., 2024), we evaluate the method on four OOD datasets, including iNaturalist (Van Horn et al., 2018), SUN (Xiao et al., 2010), Places (Zhou et al., 2017), and Textures (Cimpoi et al., 2014). Additionally, we

Table A6: Complete OOD detection results with ImageNet-1k, where a VITB/16 CLIP encoder is adopted.

| | OOD datasets | | | | | | | | | |
|---|---|---|---|---|---|---|---|---|---|---|
| Methods | INaturalist | | Sun | | Places | | Textures | | Average | |
| | AUROC↑ | FPR95↓ | AUROC↑ | FPR95↓ | AUROC↑ | FPR95↓ | AUROC↑ | FPR95↓ | AUROC↑ | FPR95↓ |
| **Methods requiring training (or fine-tuning)** | | | | | | | | | | |
| MSP (Hendrycks & Gimpel, 2016) | 87.44 | 58.36 | 79.73 | 73.72 | 79.67 | 74.41 | 79.69 | 71.93 | 81.63 | 69.61 |
| ODIN (Liang et al., 2017) | 94.65 | 30.22 | 87.17 | 54.04 | 85.54 | 55.06 | 87.85 | 51.67 | 88.80 | 47.75 |
| Energy (Liu et al., 2020) | 95.33 | 26.12 | 92.66 | 35.97 | 91.41 | 39.87 | 86.76 | 57.61 | 91.54 | 39.89 |
| GradNorm (Huang et al., 2021) | 72.56 | 81.50 | 72.86 | 82.00 | 73.70 | 80.41 | 70.26 | 79.36 | 72.35 | 80.82 |
| ViM (Wang et al., 2022) | 93.16 | 32.19 | 87.19 | 54.01 | 83.75 | 60.67 | 87.18 | 53.94 | 87.82 | 50.20 |
| KNN (Sun et al., 2022) | 94.52 | 29.17 | 92.67 | 35.62 | 91.02 | 39.61 | 85.67 | 64.35 | 90.97 | 42.19 |
| VOS (Du et al., 2022b) | 94.62 | 28.99 | 92.57 | 36.88 | 91.23 | 38.39 | 86.33 | 61.02 | 91.19 | 41.32 |
| NPOS (Tao et al., 2023) | 96.19 | 16.58 | 90.44 | 43.77 | 89.44 | 45.27 | 88.80 | 46.12 | 91.22 | 37.93 |
| ZOC (Esmaeilpour et al., 2022) | 86.09 | 87.30 | 81.20 | 81.51 | 83.39 | 73.06 | 76.46 | 98.90 | 81.79 | 85.19 |
| LSN (Nie et al., 2024) | 95.83 | 21.56 | 94.35 | 26.32 | 91.25 | 34.48 | 90.42 | 38.54 | 92.96 | 30.22 |
| CLIPN (Wang et al., 2023) | 95.27 | 23.94 | 93.93 | 26.17 | 92.28 | 33.45 | 90.93 | 40.83 | 93.10 | 31.10 |
| LoCoOp (Miyai et al., 2024) | 96.86 | 16.05 | 95.07 | 23.44 | 91.98 | 32.87 | 90.19 | 42.28 | 93.52 | 28.66 |
| LAPT (Zhang et al., 2024b) | 99.63 | 1.16 | 96.01 | 19.12 | 92.01 | 33.01 | 91.06 | 40.32 | 94.68 | 23.40 |
| NegPrompt (Li et al., 2024) | 98.73 | 6.32 | 95.55 | 22.89 | 93.34 | 27.60 | 91.60 | 35.21 | 94.81 | 23.01 |
| CMA (Kim & Hwang, 2025) | 99.62 | 1.65 | 96.36 | 16.84 | 93.11 | 27.65 | 91.64 | 33.58 | 95.13 | 19.93 |
| **Zero-Shot Training-free Methods** | | | | | | | | | | |
| MCM (Ming et al., 2022a) | 94.59 | 32.20 | 92.25 | 38.80 | 90.31 | 46.20 | 86.12 | 58.50 | 90.82 | 43.93 |
| CoVer (Zhang et al., 2024a) | 95.98 | 22.55 | 93.42 | 32.85 | 90.27 | 40.71 | 90.14 | 43.39 | 92.45 | 34.88 |
| Lee *et al.* (Lee et al., 2025) | 96.89 | 23.84 | 93.69 | 30.11 | 93.17 | 29.86 | 88.47 | 47.35 | 93.05 | 32.79 |
| EOE (Cao et al., 2024) | 97.52 | 12.29 | 95.73 | 20.04 | 92.95 | 30.16 | 85.64 | 57.53 | 92.96 | 30.09 |
| NegLabel (Jiang et al., 2024) | 99.49 | 1.91 | 95.49 | 20.53 | 91.64 | 35.59 | 90.22 | 43.56 | 94.21 | 25.40 |
| CLIPScope (Fu et al., 2024) | 99.61 | 1.29 | 96.77 | 15.56 | 93.54 | 28.45 | 91.41 | 38.37 | 95.30 | 20.88 |
| AdaNeg (Zhang & Zhang, 2024) | 99.71 | 0.59 | 97.44 | 9.50 | 94.55 | 34.34 | 94.93 | 31.27 | 96.66 | 18.92 |
| OODD (Yang et al., 2025) | 99.79 | 0.85 | 97.17 | 12.94 | 92.51 | 30.68 | 94.51 | 30.67 | 96.00 | 18.79 |
| CSP (Chen et al., 2024) | 99.60 | 1.54 | 96.66 | 13.66 | 92.90 | 29.32 | 93.86 | 25.52 | 95.76 | 17.51 |
| **AANeg (Ours)** | **99.84** | **0.42** | **99.07** | **3.53** | **95.87** | **21.90** | **97.11** | **13.38** | **97.97** | **9.81** |

Table A7: Detailed OOD detection results on the OpenOOD benchmark, where ImageNet-1k is adopted as ID dataset.

| Near/Far-OOD | OOD Datasets | FPR95 ↓ | AUROC ↑ |
|---|---|---|---|
| Near-OOD | SSB-hard (Vaze et al., 2021) | 62.51 | 83.96 |
| | NINCO (Bitterwolf et al., 2023) | 57.61 | 85.10 |
| | **Mean** | 60.06 | 84.53 |
| Far-OOD | iNaturalist (Van Horn et al., 2018) | 0.47 | 99.83 |
| | Textures (Cimpoi et al., 2014) | 11.22 | 97.53 |
| | OpenImage-O (Wang et al., 2022) | 39.95 | 91.92 |
| | **Mean** | 17.21 | 96.43 |

validate our approach on the OpenOOD benchmark (Zhang et al., 2023; Yang et al., 2022), where OOD datasets are categorized into near-OOD (e.g., SSB-hard (Vaze et al., 2021), NINCO (Bitterwolf et al., 2023)) and far-OOD (e.g., iNaturalist (Van Horn et al., 2018), Textures (Cimpoi et al., 2014), OpenImage-O (Wang et al., 2022)) based on their similarity to the ImageNet dataset. Besides the traditional OOD detection with semantic-shift, we also study a more practical full-spectrum OOD detection setting (Yang et al., 2023a), where the model is additionally challenged by the robustness to covariate shifts. Following (Zhang et al., 2023), we adopt ImageNet-C (Hendrycks & Dietterich, 2019), ImageNet-R (Hendrycks et al., 2021), and ImageNet-V2 (Recht et al., 2019) as the covariate-shifted test data in the full-spectrum setting.

### A6.3 DETAILED RESULTS ON IMAGENET DATASET

We compare against traditional methods (Hendrycks & Gimpel, 2016; Liang et al., 2017; Liu et al., 2020; Huang et al., 2021; Wang et al., 2022; Sun et al., 2022; Du et al., 2022b; Tao et al., 2023) by fine-tuning CLIP-encoders with labeled training samples, as described in (Jiang et al., 2024), and report results of (Nie et al., 2024; Miyai et al., 2024; Jiang et al., 2024; Li et al., 2024; Bai et al., 2023; Zhang et al., 2024b; Li et al., 2024; Kim & Hwang, 2025; Chen et al., 2024; Yang et al., 2025) based on their original papers.

Table A8: Detailed full-spectrum OOD detection results on the OpenOOD benchmark, where ImageNet-1k, ImageNet-C, ImageNet-R, ImageNet-V2 are used as ID datasets.

| Near/Far-OOD | OOD Datasets | FPR95 ↓ | AUROC ↑ |
|---|---|---|---|
| Near-OOD | SSB-hard (Vaze et al., 2021) | 70.50 | 79.29 |
| | NINCO (Bitterwolf et al., 2023) | 66.92 | 78.51 |
| | **Mean** | 68.71 | 78.90 |
| Far-OOD | iNaturalist (Van Horn et al., 2018) | 1.70 | 99.58 |
| | Textures (Cimpoi et al., 2014) | 19.68 | 94.81 |
| | OpenImage-O (Wang et al., 2022) | 46.07 | 88.66 |
| | **Mean** | 22.48 | 94.35 |

Table A9: OOD detection results with CIFAR10/100 on the OpenOOD benchmark. Full results are provided in Tables A10.

| Methods | FPR95 ↓ | | AUROC ↑ | |
|---|---|---|---|---|
| | Near-OOD | Far-OOD | Near-OOD | Far-OOD |
| **Methods requiring training (or fine-tuning)** | | | | |
| PixMix (Hendrycks et al., 2022) + KNN (Sun et al., 2022) | – | – | 93.10 | 95.94 |
| OE (Hendrycks et al., 2018) + MSP (Hendrycks & Gimpel, 2016) | – | – | 94.82 | 96.00 |
| PixMix (Hendrycks et al., 2022) + RotPred (Hendrycks et al., 2019a) | – | – | 94.86 | 98.18 |
| **Zero-shot Training-free Methods** | | | | |
| MCM (Ming et al., 2022a) | 30.86 | 17.99 | 91.92 | 95.54 |
| NegLabel (Jiang et al., 2024) | 28.75 | 6.60 | 94.58 | 98.39 |
| AdaNeg (Zhang & Zhang, 2024) | 20.40 | 2.79 | 94.78 | 99.26 |
| **AANeg (Ours)** | **19.31** | **2.43** | **94.91** | **99.26** |

(a) Results with ID dataset of CIFAR10

| Methods | FPR95 ↓ | | AUROC ↑ | |
|---|---|---|---|---|
| | Near-OOD | Far-OOD | Near-OOD | Far-OOD |
| **Methods requiring training (or fine-tuning)** | | | | |
| GEN (Liu et al., 2023) | – | – | 81.31 | 79.68 |
| VOS (Du et al., 2022c) + EBO (Liu et al., 2020) | – | – | 80.93 | 81.32 |
| SCALE (Xu et al., 2023) | – | – | 80.99 | 81.42 |
| OE (Hendrycks et al., 2018) + MSP (Hendrycks & Gimpel, 2016) | – | – | 88.30 | 81.41 |
| **Zero-shot Training-free Methods** | | | | |
| MCM (Ming et al., 2022a) | 75.20 | 59.32 | 71.00 | 76.00 |
| NegLabel (Jiang et al., 2024) | 71.44 | 40.92 | 70.58 | 89.68 |
| AdaNeg (Zhang & Zhang, 2024) | 59.07 | 29.35 | 84.62 | 95.25 |
| **AANeg (Ours)** | **57.74** | **24.55** | **85.06** | **95.41** |

(b) Results with ID dataset of CIFAR100.

The detailed OOD detection results with OOD datasets of INaturalist/Sun/Places/Textures are illustrated in Tab. A6.

The detailed OOD detection results and full-spectrum OOD detection results under the OpenOOD setting are presented in Tab. A7 and Tab. A8, respectively.

### A6.4 RESULTS ON CIFAR10/100

Besides ImageNet, we also assess our method on the smaller CIFAR10/100 datasets (Krizhevsky et al., 2009) under the OpenOOD framework. Specifically, with CIFAR10/100 as the ID datasets, we utilize near-OOD datasets such as CIFAR100/10 and TIN (Le & Yang, 2015), and far-OOD datasets including MNIST (Deng, 2012), SVHN (Netzer et al., 2011), Texture (Cimpoi et al., 2014), and Places365 (Zhou et al., 2017). As illustrated in Tab. A9, our advantage still holds.

### A6.5 RESULTS WITH MEDICAL IMAGES

We also validate our AANeg method with medical images following the OpenOOD setup (Zhang et al., 2023). Specifically, we use BIMCV as the ID dataset, where the task is to distinguish COVID-19 patients from healthy individuals using chest X-ray images (Vayá et al., 2020). The OOD dataset is

Table A10: Detailed OOD detection results on the OpenOOD benchmark.

| Near/Far-OOD | Datasets | FPR95 ↓ | AUROC ↑ |
|---|---|---|---|
| Near-OOD | CIFAR100 (Krizhevsky et al., 2009) | 33.63 | 91.17 |
| | TIN (Le & Yang, 2015) | 4.99 | 98.65 |
| | **Mean** | 19.31 | 94.91 |
| Far-OOD | MNIST (Deng, 2012) | 0.13 | 99.96 |
| | SVHN (Netzer et al., 2011) | 0.04 | 99.97 |
| | Texture (Cimpoi et al., 2014) | 0.04 | 99.86 |
| | Places365 (Zhou et al., 2017) | 9.51 | 97.25 |
| | **Mean** | 2.43 | 99.26 |

(a) Detailed results with ID dataset of CIFAR10.

| Near/Far-OOD | Datasets | FPR95 ↓ | AUROC ↑ |
|---|---|---|---|
| Near-OOD | CIFAR10 (Krizhevsky et al., 2009) | 56.32 | 80.47 |
| | TIN (Le & Yang, 2015) | 59.16 | 89.65 |
| | **Mean** | 57.74 | 85.06 |
| Far-OOD | MNIST (Deng, 2012) | 0.54 | 99.81 |
| | SVHN (Netzer et al., 2011) | 5.81 | 98.53 |
| | Texture (Cimpoi et al., 2014) | 31.36 | 95.36 |
| | Places365 (Zhou et al., 2017) | 60.49 | 90.94 |
| | **Mean** | 24.55 | 95.41 |

(b) Detailed results with ID dataset of CIFAR100.

Table A11: OOD detection results with medical images following the OpenOOD setting.

| Methods | OOD datasets | | | | | |
|---|---|---|---|---|---|---|
| | CT-SCAN | | X-Ray-Bone | | Average | |
| | AUROC↑ | FPR95↓ | AUROC↑ | FPR95↓ | AUROC↑ | FPR95↓ |
| NegLabel (Jiang et al., 2024) | 63.53 | 100.0 | 99.68 | 0.56 | 81.61 | 50.28 |
| AdaNeg (Zhang & Zhang, 2024) | 93.48 | 100.0 | 99.99 | 0.11 | 96.74 | 50.06 |
| **AANeg (Ours)** | **93.94** | **39.06** | **100.0** | **0.0** | **96.97** | **19.53** |

constructed using the CT-SCAN and X-Ray-Bone datasets. The CT-SCAN dataset includes computed tomography (CT) images of COVID-19 patients and healthy individuals, while the X-Ray-Bone dataset contains X-ray images of hands. As shown in Tab. A11, our method significantly outperforms existing competitors.

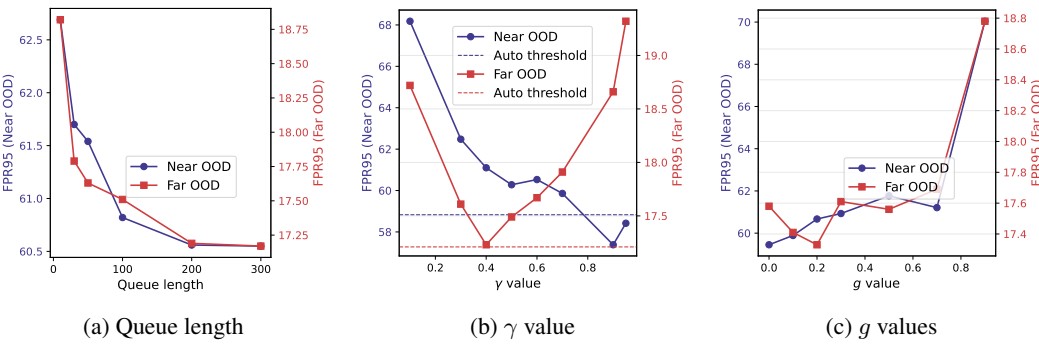

(a) Queue length      (b) $\gamma$ value      (c) $g$ values

Figure A4: Analyses on (a) the queue length $L$, (b) threshold $\gamma$, and (c) gap value $g$ in Eq. 9 under the OpenOOD setup.

A6.6 MORE ANALYSES

**Queue length.** We formulate $\mathcal{X}_{pos/neg}$ as fixed-length FIFO queues, where the queue length $L$ determines the number of cached samples in dynamic environments. As shown in Fig. A4a, appropriately increasing the queue length can reduce error, and performance saturates when the queue length exceeds 300. Therefore, we set the queue length to 300 by default.

$\gamma$ **values.** As shown in Fig. A4b, the optimal threshold $\gamma$ differs between near-OOD and far-OOD scenarios because the distance between OOD samples and ID samples varies significantly. Instead of manually searching for the best $\gamma$, we adopt an automated approach to set it dynamically. This is achieved by modeling the score $\mathcal{S}_{aa}$ of all historical samples as a bimodal distribution of two clusters (*e.g.*, ID Vs. OOD) and identifying the threshold that minimizes the intra-cluster variations of the two clusters (Li et al., 2023):

$$\min_{\gamma} \quad \mathrm{var}(\mathcal{O}_{\mathrm{his}}^{+}) + \mathrm{var}(\mathcal{O}_{\mathrm{his}}^{-}) \tag{A6.1}$$

$$\text{where} \quad \mathcal{O}_{\mathrm{his}}^{+} = \{\mathcal{S}_{aa}(\boldsymbol{v}) \mid \mathcal{S}_{aa}(\boldsymbol{v}) \geq \gamma, \ \boldsymbol{v} \in \mathcal{X}_{\mathrm{his}}\}, \tag{A6.2}$$

$$\mathcal{O}_{\mathrm{his}}^{-} = \{\mathcal{S}_{aa}(\boldsymbol{v}) \mid \mathcal{S}_{aa}(\boldsymbol{v}) < \gamma, \ \boldsymbol{v} \in \mathcal{X}_{\mathrm{his}}\}, \tag{A6.3}$$

where $\mathrm{var}(A)$ measures the variation of the score set $A$, and $\mathcal{X}_{\mathrm{his}}$ is also a FIFO queue that stores the most recent $20,000$ test samples, initialized with positive and negative samples as defined in Eq. 10. This dynamic thresholding typically achieves results comparable to those with manually searched $\gamma$ and is adopted as the default setting.

$g$ **values.** As shown in Fig. A4c, an appropriately small gap $g$ helps reduce the detection error in the Far OOD setting, whereas the Near OOD setting benefits from a smaller value of $g$. By default, we set $g = 0.2$ for all experiments.

**VLM Backbones.** As shown in Tab. A12, our AANet achieves excellent results across various VLM backbones, where a stronger backbone generally leads to better performance. Interestingly, unlike NegLabel, which heavily relies on a strong backbone to achieve high results, our method works well even with a smaller ResNet-50 backbone, demonstrating its effectiveness and practicality.

Table A12: OOD detection results of AANeg with different VLMs backbones, where ImageNet-1K is used as the ID dataset.

| Backbone | OOD datasets | | | | | | | | | |
|---|---|---|---|---|---|---|---|---|---|---|
| | INaturalist | | Sun | | Places | | Textures | | Average | |
| | AUROC↑ | FPR95↓ | AUROC↑ | FPR95↓ | AUROC↑ | FPR95↓ | AUROC↑ | FPR95↓ | AUROC↑ | FPR95↓ |
| | **ResNet50** | | | | | | | | | |
| NegLabel (Jiang et al., 2024) | 99.24 | 2.88 | 94.54 | 26.51 | 89.72 | 42.60 | 88.40 | 50.80 | 92.97 | 30.70 |
| **AANeg (Ours)** | **99.73** | **0.99** | **99.04** | **4.00** | **95.97** | **23.76** | **97.01** | **12.71** | **97.88** | **10.37** |
| | **VITB/32** | | | | | | | | | |
| NegLabel (Jiang et al., 2024) | 99.11 | 3.73 | 95.27 | 22.48 | 91.72 | 34.94 | 88.57 | 50.51 | 93.67 | 27.92 |
| **AANeg (Ours)** | **99.76** | **0.87** | **99.24** | **3.09** | **96.07** | **20.97** | **96.50** | **16.28** | **97.89** | **10.30** |
| | **VITB/16** | | | | | | | | | |
| NegLabel (Jiang et al., 2024) | 99.49 | 1.91 | 95.49 | 20.53 | 91.64 | 35.59 | 90.22 | 43.56 | 94.21 | 25.40 |
| **AANeg (Ours)** | **99.84** | **0.42** | **99.07** | **3.53** | **95.87** | **21.90** | **97.11** | **13.38** | **97.97** | **9.81** |
| | **VITL/14** | | | | | | | | | |
| NegLabel (Jiang et al., 2024) | 99.53 | 1.77 | 95.63 | 22.33 | 93.01 | 32.22 | 89.71 | 42.92 | 94.47 | 24.81 |
| **AANeg (Ours)** | **99.88** | **0.29** | **99.15** | **3.42** | **96.11** | **20.79** | **97.12** | **13.57** | **98.07** | **9.52** |

Table A13: OOD detection results with different corpus datasets on the OpenOOD benchmark, where ImageNet-1k is adopted as ID dataset.

| Corpus Datasets | FPR95 ↓ | | AUROC ↑ | |
|---|---|---|---|---|
| | Near-OOD | Far-OOD | Near-OOD | Far-OOD |
| Vanilla WordNet | 60.52 | 17.34 | 83.24 | 95.92 |
| Part-of-speech-tagging | 60.36 | 17.24 | 83.74 | 96.15 |
| WordNet-subset Filtered by NegLabel | 60.06 | 17.21 | 84.53 | 96.43 |

**Corpus Datasets.** The corpus dataset provides candidates for activated labels, and its diversity is a prerequisite for selecting effective activated labels. We construct the corpus dataset by selecting

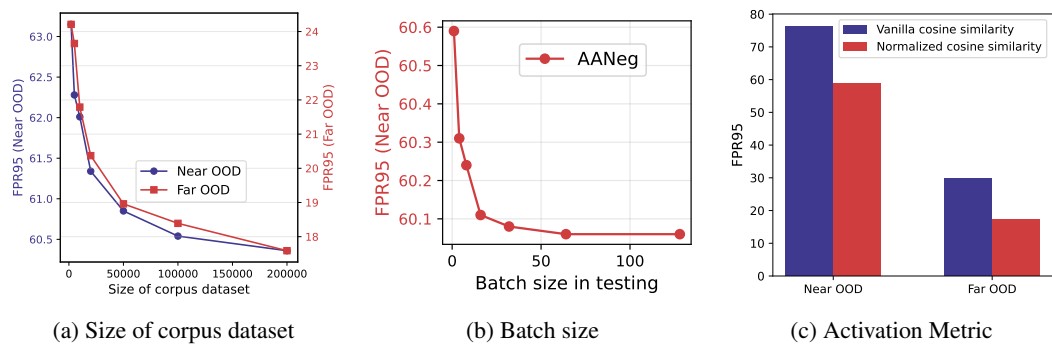

(a) Size of corpus dataset       (b) Batch size       (c) Activation Metric

Figure A5: Analyses on (a) size of corpus dataset, (b) the batch size, and (c) activation metric under the OpenOOD setup.

nouns and adjectives from WordNet [1] and Part-of-Speech-Tags [2], removing duplicate words within the ID set. This results in corpus datasets containing 140.5K and 270.2K words, respectively. For a fair comparison, we also adopt the WordNet subset filtered by NegLabel as the corpus dataset, which contains 70K adjectives and nouns.

As illustrated in Tab. A13, different corpus datasets lead to similarly good results, demonstrating the robustness of our method to the choice of corpus dataset. Beyond comparing different corpus datasets, we also analyze the sensitivity of our method to the size of the corpus dataset by randomly sampling smaller subsets from the Part-of-Speech-Tags dataset. As shown in Fig. A5a, using an appropriately large-sized corpus dataset generally results in better performance, with results saturating when the corpus dataset exceeds 100K. For fair comparison, we adopt the WordNet subset filtered by NegLabel as the corpus dataset by default.

**Batch Size.** Our method leverages batch information to obtain batch-adaptive activated labels; thus, it is necessary to explore its sensitivity to batch size. As shown in Fig. A5b, a larger batch size generally performs better. Particularly, in the case of batch size = 1, it approaches the distribution-adaptive variants and still significantly outperforms the NegLabel competitor.

**Activation Metric.** The activation metric we use is the cosine similarity normalized by a softmax function, as shown in Eq. 5. One may wonder whether directly using vanilla cosine similarity, *i.e.*, modifying Eq. 5 to $Act(\mathcal{X}, \widehat{y}_i) = \frac{1}{|\mathcal{X}|} \sum_{\boldsymbol{x} \in \mathcal{X}} f_{img}(\boldsymbol{x}) f_{txt}(\rho(\widehat{y}_i))$, could also work. As shown in Fig. A5c, using the normalized similarity leads to a significant advantage. This is likely because unnormalized similarity has a lower discriminative capacity. For example, same-class similarities often cluster around 0.3, while different-class similarities typically cluster around 0.1 (Ming et al., 2022a), making it difficult to distinguish activated labels effectively.

**Statistical Significance.** We control the input sequence using a random seed and report OOD detection results on the ImageNet-1k dataset with three random seeds as follows: AUROC: $97.97 \pm 0.01$ and FPR95: $9.81 \pm 0.01$. These results validate the robustness of our method to variations in the input sequence.

**Mutual Enhancement.** In our method, improving the accuracy of the activation score in Eq. 8 helps select more effective negative labels in Eq. 6, thereby enhancing the OOD detection capability of the activation-aware score in Eq. 15. Similarly, the improved OOD detection capability of the activation-aware score contributes to selecting more accurate negative and positive images in Eq. 9, which in turn enhances the estimation of the activation score in Eq. 8. Overall, there exists a mutual enhancement relationship between the estimation of the activation score and the activation-aware OOD score function. Such a relationship is evidenced in Tab. A14, where we remove this mutual enhancement by fixing $\mathcal{S}_{aa}$ in Eq. 9 using activated labels estimated via the initialized $\mathcal{X}_{pos/neg}$ in Eq. 10. This leads to a significant performance drop.

---

[1] https://wordnet.princeton.edu/
[2] https://www.kaggle.com/datasets/ruchi798/part-of-speech-tagging

Table A14: Analyses on the mutual enhancement between the estimation of activation information and the activation-aware OOD score function.

| Methods | FPR95 ↓ | |
| --- | --- | --- |
| | Near-OOD | Far-OOD |
| W/o mutual enhancement | 63.00 | 19.04 |
| With mutual enhancement (Ours) | 60.06 | 17.21 |

Table A15: OOD detection results under the temporal shifts, where ImageNet-1k ID dataset and a VITB/16 CLIP encoder are adopted.

| Methods | OOD datasets | | | | | | | | | |
| --- | --- | --- | --- | --- | --- | --- | --- | --- | --- | --- |
| | INaturalist | | Sun | | Places | | Textures | | Average | |
| | AUROC↑ | FPR95↓ | AUROC↑ | FPR95↓ | AUROC↑ | FPR95↓ | AUROC↑ | FPR95↓ | AUROC↑ | FPR95↓ |
| NegLabel (Jiang et al., 2024) | 99.49 | 1.91 | 95.49 | 20.53 | 91.64 | 35.59 | 90.22 | 43.56 | 94.21 | 25.40 |
| **AANeg under Temporal Shifts** | | | | | | | | | | |
| I-S-P-T | 99.84 | 0.45 | 98.77 | 4.75 | 95.93 | 22.16 | 96.67 | 14.91 | 97.80 | 10.56 |
| S-P-T-I | 99.83 | 0.46 | 99.04 | 3.65 | 95.92 | 21.37 | 96.70 | 15.09 | 97.87 | 10.14 |
| P-T-I-S | 99.84 | 0.45 | 98.73 | 5.28 | 95.87 | 21.67 | 96.66 | 15.33 | 97.87 | 11.68 |
| T-I-S-P | 99.83 | 0.47 | 98.77 | 4.69 | 95.94 | 21.30 | 97.04 | 13.51 | 97.89 | 9.99 |

**Temporal Shift.** Since our method dynamically explores activated negative labels in a test-time adaptation manner, it is crucial to investigate its stability under temporal shifts, where OOD environments evolve over time. Specifically, we use ImageNet as the ID dataset and assume that OOD datasets change sequentially over time (*e.g.*, I-S-P-T represents OOD datasets transitioning from INaturalist to Sun to Places to Textures). In implementation, we do not re-initialize the queue $\mathcal{X}_{pos/neg}$ for each OOD dataset. As shown in Tab. A15, our method consistently maintains a large advantage over the NegLabel baseline, validating its robustness to temporal shifts.

**Sample Order.** In our experiments, ID and OOD samples are randomly shuffled during testing, corresponding to the "Random Shuffled" scenario. To analyze the impact of sample order, we have conducted additional experiments. Specifically, we consider two extreme cases: "ID First" (all ID samples are tested before any OOD samples) and "OOD First" (all OOD samples are tested before any ID samples). As shown in Tab. A16, while performance does drop under these extreme settings compared to the "Random Shuffled" scenario, our method still significantly outperforms the NegLabel baseline. This demonstrates the robustness and effectiveness of our method, even when the order of test samples is highly imbalanced.

Table A16: FPR95 (↓) with different orders of ID and OOD samples. .

| Order of Test Samples | INaturalist | SUN | Places | Textures | Average |
| --- | --- | --- | --- | --- | --- |
| NegLabel (Baseline) | 1.91 | 20.53 | 35.59 | 43.56 | 25.40 |
| ID First | 2.97 | 8.35 | 29.63 | 24.91 | 16.47 |
| OOD First | 2.74 | 9.06 | 36.00 | 19.14 | 16.73 |
| Random Shuffled | 0.42 | 3.53 | 21.90 | 13.38 | 9.81 |

**Limitations.** The limitation of our method lies in the assumption that the corpus dataset covers words related to the OOD distribution and that the pre-trained text encoder understands these words. This assumption may not hold in certain domains. For example, WordNet primarily contains everyday vocabulary and lacks sufficient medical terms, while the vanilla CLIP model has limited understanding of medical images. As a result, improvements in medical OOD detection can be restricted. This limitation suggests that domain-specific tasks may require a tailored corpus dataset and pre-trained models, which is left for future investigation.

**Use of Large Language Model.** We utilize language models to enhance English writing by improving logical coherence, checking word usage, and identifying typos.

**Visualization of Activated Labels.** We visualize the activated labels under different OOD datasets and activation criteria in Fig. A6. We observe that the selected activated labels indeed exhibit a

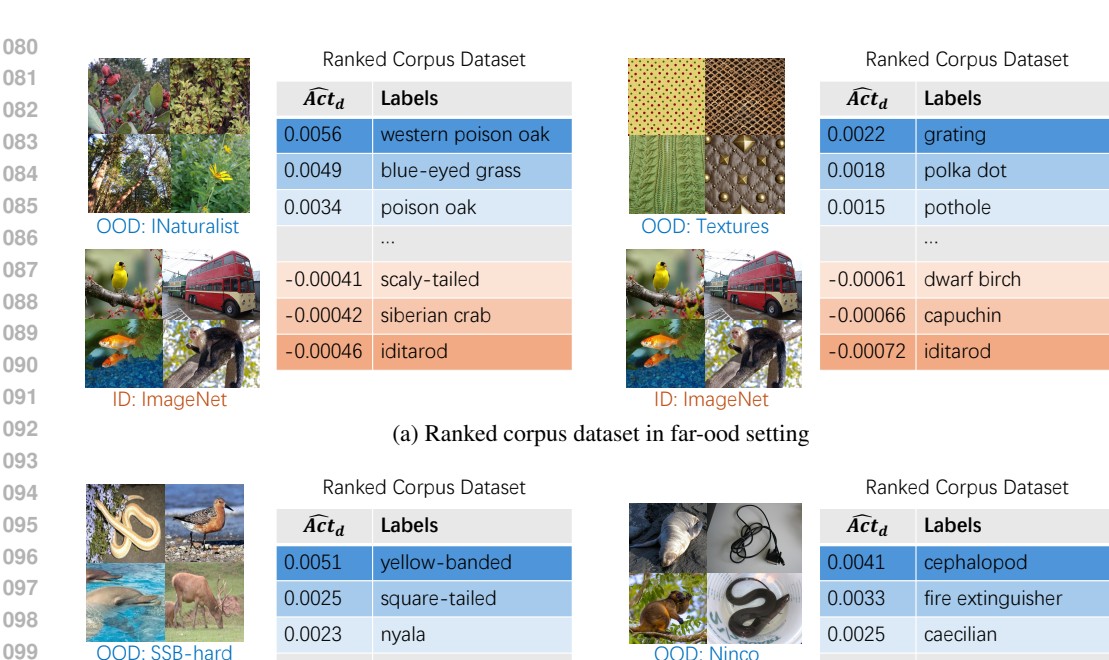

(a) Ranked corpus dataset in far-ood setting

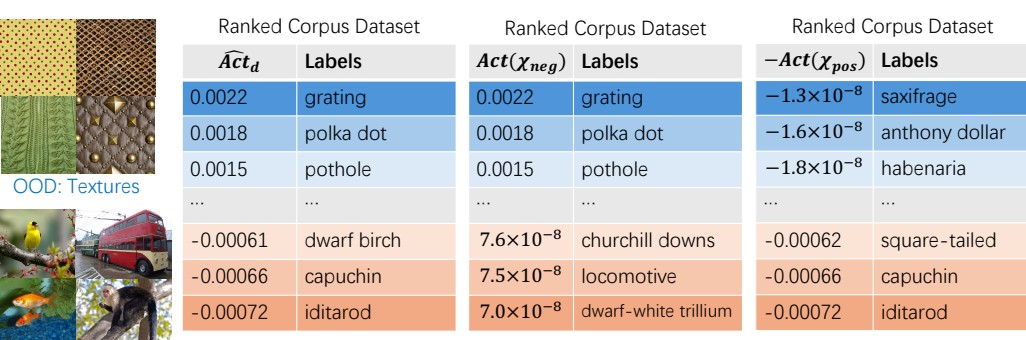

(b) Ranked corpus dataset in near-ood setting

| $\widehat{Act_d}$ | Labels | $Act(\mathcal{X}_{neg})$ | Labels | $-Act(\mathcal{X}_{pos})$ | Labels |
|---|---|---|---|---|---|
| 0.0022 | grating | 0.0022 | grating | $-1.3\times10^{-8}$ | saxifrage |
| 0.0018 | polka dot | 0.0018 | polka dot | $-1.6\times10^{-8}$ | anthony dollar |
| 0.0015 | pothole | 0.0015 | pothole | $-1.8\times10^{-8}$ | habenaria |
| ... | ... | ... | ... | ... | ... |
| -0.00061 | dwarf birch | $7.6\times10^{-8}$ | churchill downs | -0.00062 | square-tailed |
| -0.00066 | capuchin | $7.5\times10^{-8}$ | locomotive | -0.00066 | capuchin |
| -0.00072 | iditarod | $7.0\times10^{-8}$ | dwarf-white trillium | -0.00072 | iditarod |

(c) Ranked corpus dataset with different ranking criteria

Figure A6: Visualization of the ranked corpus dataset with (a) far-ood setting, (b) near-ood setting, and (c) different ranking criteria.

high degree of similarity with the OOD dataset. For example, when the OOD dataset is *Textures*, the labels with high activation scores are 'grating' and 'polka dot', which align well with the visual characteristics of the *Textures* dataset, as shown in Fig. A6a. Furthermore, we find that the highly activated scores selected using $Act(\mathcal{X}_{neg})$ and $\widehat{Act_d}$ are quite similar, which explains the effectiveness of $Act(\mathcal{X}_{neg})$ as a criterion, as demonstrated in Fig. 3b. In contrast, the activated labels selected using only positive images do not show a clear relationship with the target OOD dataset, which corresponds to its lower results in Fig. 3b..

