# OpenReview forum: "Activation Matters: Adaptive and Activated Negative Labels for OOD Detection with Vision-Language Models"
_ICLR.cc/2026/Conference — ICLR 2026 Conference Withdrawn Submission_

### Official Review · Reviewer_vr2d · 2025-10-24

**Soundness:** 3
**Presentation:** 3
**Contribution:** 3
**Rating:** 6
**Confidence:** 4

**Summary:**

This paper proposes an Adaptive Activated Negative Label–guided (AANeg) method for OOD detection. AANeg dynamically selects negative labels that are strongly activated by OOD samples but weakly activated by ID samples, and further introduces an activation-aware scoring function that emphasizes more informative negatives. As a zero-shot, training-free approach, AANeg achieves state-of-the-art performance on large-scale benchmarks—for example, on ImageNet it reduces FPR95 from 17.5% to 9.8%—with both theoretical analysis and extensive experiments supporting its design.

**Strengths:**

1. AANeg leverages an explicit activation metric to dynamically screen negative labels that exhibit high activation on OOD inputs and low activation on ID inputs. This mitigates the issue in prior NegLabel-style approaches where many OOD labels have low activation and limited discriminative value, thereby improving separability.

2.  On large-scale benchmarks (e.g., ImageNet as ID), AANeg reduces FPR95 from 17.5% to 9.8% and shows stable performance across diverse settings, including Near-OOD, full-spectrum OOD, and medical OOD. The paper also provides theoretical justifications that align with the empirical findings.

**Weaknesses:**

1.  AANeg assumes the corpus (and text encoder) contains vocabulary relevant to the target OOD distribution. In specialized domains (e.g., healthcare), insufficient domain terminology may limit the method’s ability to capture OOD characteristics, constraining performance.

2. The queue initialization relies on ID label features and random noise images, introducing two potential issues:
   &#x20;(a) In few-shot ID scenarios (e.g., only tens or hundreds of ID samples), ID label features may lack representativeness, biasing the initial activation estimates and subsequent negative-label selection.
   &#x20;(b) Random noise differs markedly from real OOD distributions, so the initial negative queue may not reflect true OOD structure, potentially causing unstable detection in the early testing phase (first several batches). The paper does not evaluate this specifically under few-shot ID conditions.

**Questions:**

1. Although AANeg is validated across multiple OOD regimes, it would be valuable to evaluate robustness under extremely heterogeneous OOD (e.g., fully cross-modal inputs). Even if performance degrades, such results would offer actionable insights for future extensions.

2. Real-world OOD data often include blur, occlusion, and annotation noise. If OOD samples are severely degraded, will the activation metric be disrupted? For instance, would blurry texture images substantially suppress activations for labels like “grating,” inadvertently filtering out otherwise effective negatives?

3. The batch-adaptive variant (Sec. 3.3.2) adjusts activation scores using positives/negatives within the current batch but does not explicitly address extreme batch compositions—e.g., a batch of all-OOD samples or >90% OOD. As shown in Appendix A6.6 (Table A16), when the testing order is OOD-first, the average FPR95 increases from 9.81% to 16.73%. Further experiments (or safeguards) for such adversarial testing orders would strengthen the claims.

---

### Official Review · Reviewer_PohU · 2025-10-31

**Soundness:** 2
**Presentation:** 2
**Contribution:** 2
**Rating:** 2
**Confidence:** 4

**Summary:**

this paper dynamically evaluates activation levels across the corpus dataset and selects words with high activation responses as negative labels.  On the large-scale ImageNet benchmark, AANeg achieves the SOTA performance

**Strengths:**

Generally clear and readable; figures and tables are informative;

**Weaknesses:**

1. This paper frames the activation-aware score $S_{\mathrm{aa}}$ as mutual enhancement but provides no stability or robustness guarantees. Early mis-queues could poison subsequent selection.
2. The derivative analysis states improvement when $p_1 - p_2 < 0$, which is tautological. There is no treatment of estimation error from FIFO queues or conditions guaranteeing $p_1 - p_2 < 0$ after the approximate $\widehat{\mathrm{Act}}$. Provide at least a concentration bound for $\widehat{\mathrm{Act}}$ under streaming noise and a basic stability result.
3.what is the size of the Image batch

**Questions:**

see weakness

---

### Official Review · Reviewer_5eoA · 2025-10-31

**Soundness:** 2
**Presentation:** 2
**Contribution:** 2
**Rating:** 4
**Confidence:** 4

**Summary:**

This paper proposes AANeg, a method for out-of-distribution (OOD) detection using pre-trained vision-language models. The core idea is to improve upon existing negative-label-based approaches by dynamically selecting more effective negative labels during test time. The authors introduce a label "activation" metric to quantify the relevance of candidate negative labels to the current test data. This is achieved by maintaining FIFO queues of high-confidence in-distribution and out-of-distribution samples to approximate the test distribution. Based on this, the method adaptively selects negative labels that have high activation on OOD samples and low activation on in-distribution samples. Furthermore, a novel activation-aware scoring function is proposed to give more weight to highly activated negative labels. The method is zero-shot and training-free. Experiments on ImageNet and other benchmarks show that AANeg significantly outperforms existing methods.

**Strengths:**

1. The paper addresses an important and practical problem in OOD detection: the negative labels selected based on a static, generic corpus may not be optimal for a specific OOD test set. The motivation to adapt the negative labels to the test distribution is well-founded and intuitive.

2. The proposed approach of using memory queues of confident test samples to approximate the test distribution online is a sensible strategy for test-time adaptation.

3. The experimental results are strong. The method achieves state-of-the-art performance on several challenging OOD detection benchmarks, including ImageNet, significantly reducing the FPR95 metric compared to previous work.

**Weaknesses:**

1. The novelty of the core technical contribution, the "label activation score," is questionable. The proposed activation score in Equation 5 is defined as the average classification probability of a label over a set of images. The probability itself is calculated using the standard softmax over similarities between the image and the full set of ID and negative labels. This formulation is fundamentally the same as the similarity metric used in the baseline method, NegLabel (as seen in their scoring function, Equation 4). Re-labeling an existing similarity metric as "activation" and presenting its aggregation (averaging) as a primary contribution seems to be an overstatement of novelty. The terms "activation" and "similarity" have distinct meanings in the field, and this choice of terminology is confusing.

2. The paper suffers from inconsistencies in its technical presentation. The activation-aware score S_aa is a central component of the method. However, the formula presented in Equation 15 is different from the one illustrated in the framework diagram in Figure 2. Equation 15 describes a score based on a cumulative sum in the denominator, designed to amplify the weights of top-ranked labels. In contrast, Figure 2 depicts a simple averaging of two softmax-based scores. This discrepancy makes it difficult to understand the exact formulation and implementation of the proposed scoring function.

3. The motivation and superiority of the proposed activation-aware score (Equation 15) are not well-established. The authors claim (Lines 302-305) that their formulation implicitly assigns higher weights to labels with stronger activation. While this may be true, this effect could likely be achieved through a simpler and more direct re-weighting of the negative labels in the standard OOD score function (e.g., Equation 4).

4. The claim of "robustness to the number of negative labels" is overclaimed. The authors repeatedly highlight this as a key advantage. However, Figure 3a shows that the performance of AANeg, while degrading more gracefully than NegLabel, still clearly declines as the number of negative labels M increases. True robustness would imply stable performance across a wide range of M. The results merely show less sensitivity, which is a weaker claim.

5. The experimental evaluation framework is flawed and incomplete. The paper's method is fundamentally a test-time adaptation (TTA) approach. However, in Table 1, the authors group their method with other "Zero-Shot Training-free Methods," failing to distinguish between non-adaptive methods (like MCM, NegLabel) and other TTA methods (like AdaNeg, OODD). This categorization is misleading and obscures a fair comparison. A proper evaluation should structure the comparison into distinct categories: Training-based, Training-free & non-adaptive, and Test-time adaptation methods. More critically, the paper omits a highly relevant and recent baseline in the TTA-OOD space: Cao et al., "Noisy test-time adaptation in vision-language models," ICLR 2025.

6. The method's test-time adaptation process is highly sensitive to the initial batch of test samples, suffering from a classic "chicken-and-egg" problem. The success of the adaptation heavily relies on the accuracy of the initial classifications used to populate the memory queues (X_pos and X_neg). If the model encounters difficult or ambiguous samples early in the test stream, initial misclassifications can poison these queues. This contamination would directly corrupt the subsequent calculation of "activation scores," leading to the selection of ineffective negative labels and degrading the performance of the scoring function S_aa itself. This creates a significant risk of a vicious cycle of error accumulation, where a poorly adapted model makes more mistakes, further contaminating the queues. The paper fails to analyze or experimentally validate the method's robustness against this potential failure mode (e.g., by evaluating performance with deliberately ordered or noisy initial samples), casting doubt on its reliability in practical, non-stationary deployment scenarios.

7. The claim of the method being "training-free" is misleading, as it introduces a multitude of new, sensitive hyperparameters that require careful tuning at test time. These include the queue length L, the confidence gap g, the batch-history interpolation weight α, and the parameters governing the dynamic threshold γ. This approach does not eliminate the need for tuning but merely shifts the burden from the training phase to the deployment phase, which can be equally challenging in practice. This contradicts the core philosophy of a truly zero-shot or "plug-and-play" method, which should be applicable with minimal task-specific adjustments. The paper provides no general strategy for setting these hyperparameters, whose optimal values are likely dependent on the specific characteristics of the OOD distribution. This significantly undermines the method's claimed ease-of-use and practical applicability.

8. The paper's theoretical analysis is superficial and fails to provide justification for its core novel components. The provided analysis in Section 3.3 is merely a recapitulation of the work from NegLabel, arguing that selecting negative labels with a lower probability on ID images than on OOD images is beneficial. This is a foundational but not a novel insight. Crucially, the paper offers no theoretical grounding for its main contributions: the dynamics of the test-time adaptation mechanism and the unique mathematical formulation of the activation-aware score in Equation 15. For instance, there is no analysis of the stability or convergence properties of the FIFO queue update process. Furthermore, no theoretical justification is provided for why the specific "cumulative sum" denominator in the scoring function is a principled or optimal choice compared to other potential re-weighting schemes. Consequently, the proposed framework appears to be a collection of well-engineered heuristics rather than a principled method, leaving the reasons for its empirical success poorly understood from a theoretical standpoint.

**Questions:**

None

---

### Official Review · Reviewer_sN3M · 2025-10-31

**Soundness:** 3
**Presentation:** 3
**Contribution:** 3
**Rating:** 6
**Confidence:** 4

**Summary:**

This paper addresses out-of-distribution (OOD) detection by improving how negative labels are used. Existing methods rely on fixed, distant labels that often fail to activate on true OOD samples. The authors propose AANeg — a zero-shot, training-free approach that dynamically selects negative labels with strong activation responses. AANeg builds a label activation metric by accumulating activation statistics from high-confidence test samples, enabling adaptive alignment with the test distribution. A batch-level variant further refines this adaptation, and an activation-aware score function emphasizes highly activated negatives for better robustness and accuracy. Overall, the method is efficient, scalable, and theoretically grounded, offering a fresh perspective on adaptive OOD detection.

**Strengths:**

1. The paper proposes a novel and well-motivated approach to OOD detection by introducing an activation-guided adaptive negative label selection strategy (AANeg), which effectively alleviates the issue of low-activation or noisy negative labels observed in prior methods.

2. It demonstrates strong empirical performance, achieving a 15.6% reduction in FPR95 compared to NegLabel and outperforming the current state-of-the-art by 7.7% on ImageNet, validating the method’s effectiveness.

**Weaknesses:**

1. The method is somewhat sensitive to initialization, as it depends on ID and noise samples to initialize activation scores, which may affect early-stage stability and adaptation.

2. The authors’ claim that the method is “zero-shot, training-free, test-efficient, highly scalable, and theoretically grounded” appears somewhat overstated, since prior methods such as NegLabel and CSP could be similarly described.

3. Although the method exhibits notable performance improvements, the authors have not released the code and only provided future assurances, which undermines reproducibility and independent verification.

**Questions:**

Please refer to the Weakness part.

---

### Note · Authors · 2025-11-14

I have read and agree with the venue's withdrawal policy on behalf of myself and my co-authors.